# Evaluation of Color Change and Biodeterioration Resistance of Gewang (*Corypha utan* Lamk.) Wood

**Dodi Nandika [1],\*, Wayan Darmawan [1], Lina Karlinasari [1], Yusuf Sudo Hadi [1],
Imam Busyra Abdillah [1] and Salim Hiziroglu [2]**

1   Forest Products Department, IPB University (Bogor Agricultural University), Kampus IPB, Darmaga,
    Bogor 16680, Indonesia; wayandar@indo.net.id (W.D.); karlinasari@apps.ipb.ac.id (L.K.);
    yshadi@indo.net.id (Y.S.H.); imam_busyra@apps.ipb.ac.id (I.B.A.)
2   Natural Resource Ecology and Management, Oklahoma State University, Stillwater, OK 74078, USA;
    salim.hiziroglu@oksate.edu
\*   Correspondence: dodina@apps.ipb.ac.id

**Abstract:** Gewang (*Corypha utan* Lamk.) is one of the endemic palm species which has been used as a building material for many years in Indonesia. The objective of this study was to enhance the overall resistance of gewang wood to biological deterioration by using smoke treatment. Samples taken from different parts of the trunks, namely bottom, middle, and upper parts in a longitudinal direction and outer and inner parts in a transversal direction, were exposed to the smoking process. Discoloration, dry-wood termite (*Cryptotemes cynocephalus*) and fungi (*Schizophyllum commune*) resistance of smoked samples were determined according to the Indonesian standard. Based on the findings in this work, discoloration of smoked specimens was more prominent than that of the unsmoked samples. Overall termite and decay resistance of smoked samples were enhanced and higher than those of unsmoked samples without any influence of longitudinal and transversal orientations within the trunks. It appears that smoking can be considered as a potential method to improve decay and termite resistances of gewang wood.

**Keywords:** gewang palm wood; decay fungus; dry-wood termite damage; smoke treatment

## 1. Introduction

Log production for the Indonesian wood industry reached 47.9 million $m^3$, with 85% of production being from plantation forests in 2018 [1]. The need for timber supply in the country significantly increased by 4 million $m^3$/year, causing a serious shortage of high-quality timber in the last four years. This requires multiple approaches, including exploration of "new timber" resources, particularly from endemic species. One of the endemic tree species that grow in Nusa Tenggara Province is gewang (*Corypha utan* Lamk.) [2,3].

At a mature stage, gewang trunk could have 2.8 tons of biomass with an average density of $0.50 \text{ g} \cdot cm^{-3}$. In terms of volume, such amount of lignocellulose material has a great potential as a source of building materials. People in East Nusa Tenggara Province have been using gewang wood as building materials for a long time. The outer part of the gewang trunk is also used for flooring to replace wooden parquet [4]. However, the inner part of the gewang trunk has significantly lower density, as well as modulus of elasticity (MOE) and modulus of rupture (MOR), compared to those of the middle and outer parts [5]. This leads to ineffective utilization of the inner part of gewang trunk as a building material. The inner part of gewang trunk is also very susceptible to biodeterioration [6]. It was reported that the damage due to biodeterioration in Indonesia, especially termite attacks, reaches approximately USD 1 billion on wood-based material in buildings [7]. It is expected that this value

will increase in the future if non-durable timber continues to be used in the building constructions without any particular treatment to increase their overall resistance.

One of the most effective treatments to improve the durability of wood is smoke treatment. Such treatment has long been used in the food industry to extend product shelf life, provide flavor, and change the color of the product [8]. Smoke from wood contains a large number of polycyclic aromatic hydrocarbons, which are predominantly phenols, aldehydes, ketones, organic acids, alcohols, esters, hydrocarbons, and various heterocyclic compounds [9]. Hadi et al. [10,11] determined solid wood and glued-laminated lumber specimens exposed to wood smoke are more resistant to subterranean termite attack than control specimens, and two weeks exposure time to smoke increases the resistance of sengon (*Falcataria moluccana*), manii (*Maesopsis eminii*) and mangium (*Acacia mangium*) also as a result of smoke treatment resistance level of the sample increased one class based on the Indonesian standard. However, longer smoking periods are required to reach the higher class (or very resistant) level of resistance. Smoke treatment with a longer period lead to color changes in sengon and mangium wood compared with untreated wood [12]. There is very little or no information on color change and biodeterioration resistance of smoked gewang wood as a function of smoke treatment.

Therefore, the main objective of this study was to determine the color change and resistance of smoked gewang wood specimens from different parts of the trunk, longitudinally as well as transversally, to dry-wood termite and decay fungus attacks when they were exposed for three weeks to function smoke treatment. The resistance of the control and smoked treated wood specimens were also compared.

## 2. Materials and Methods

Three gewang (*C. utan*) logs, 1.30 m in length and 30–40 cm in diameter, were obtained from Kupang, East Nusa Tenggara, Indonesia. Each log had a longitudinal orientation of the trunk, namely bottom part (B), middle part (M), and upper part (U). The logs were cut from two gewang trees that had grown in a lowland community forest at the age of about 30 years (mature tree). After reaching air-dry condition, three disks with a thickness of 15 cm were then prepared from each log, namely Bf, Bt, and Bd (represent bottom part); Mf, Mt, and Md (represent middle part); and Uf, Ut, and Ud (represent upper part) as shown in Figure 1.

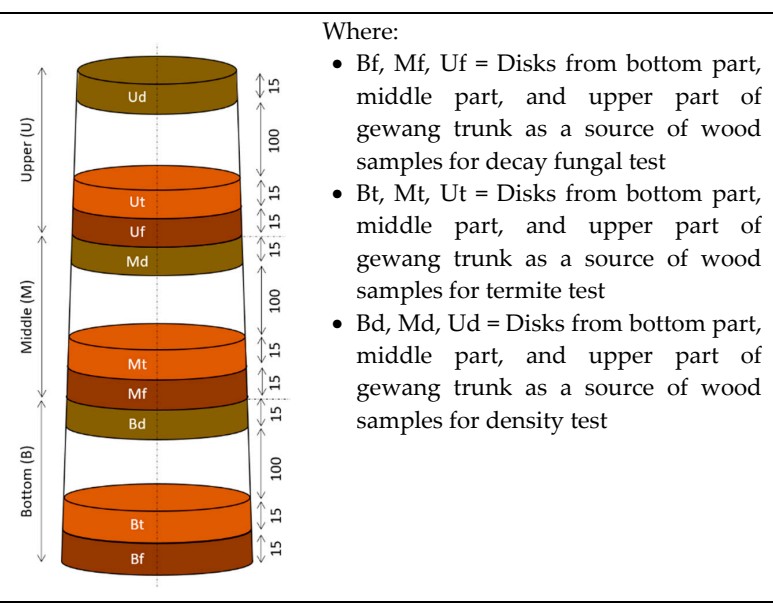

**Figure 1.** Cutting pattern of gewang log to prepare wood specimens representing bottom part (**B**), middle part (**M**), and upper part (**U**) of the trunk.



Bf, Mf, and Uf disks were prepared for decay resistance to wood samples; Bt, Mt, and Ut disks were prepared for termite resistance wood samples; and Bd, Md, and Ud disks were prepared for density test respectively. Each disk was processed for flat-shown lumbers, then cut into wood specimens of 2.5 cm by 2.5 cm in a cross-section along 1 m of the trunk length (Figure 2).

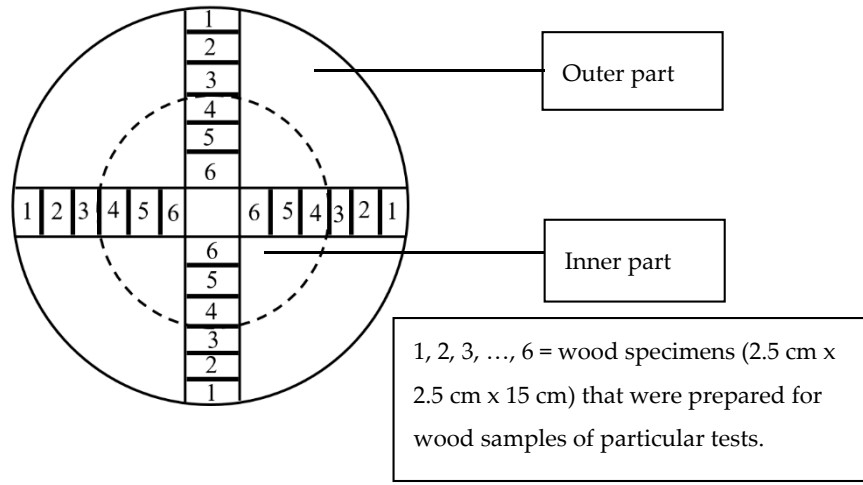

**Figure 2.** Cutting pattern of each gewang discs to prepare wood sample, representing outer part and inner part of gewang trunk.

### 2.1. Smoking Procedure

Kesambi (*Schleichera oleosa*) wood was pyrolyzed to produce charcoal, and the smoke released as a byproduct was used for the smoking process, for three weeks. The smoke produced from pyrolyzed tank flew to another tank for catching the tar and reducing the temperature and then flew to the smoking chamber. The position of wood samples in the chamber was set up randomly. The temperature inside the chamber during the smoking process ranged from 18 to 36 °C, and fluctuated in line with the day and night ambient temperature. After completion of the smoking process, the samples then underwent conditioning for 2 weeks under room temperature of 26 °C (min 20.1, max 31.1 °C) and average relative humidity of 83.4% (range, 62.5–95.6%). Five replications were considered for each type of sample.

### 2.2. Density Measurement of The Samples

Wood density was determined through measuring volume of wood specimen at air-dry condition ($V_1$), and oven-dry weight ($W_1$), and the wood density was calculated as follows:

$$\text{Density} \left(\text{g·cm}^{-3}\right) = W_1/V_1 \tag{1}$$

### 2.3. Discoloration Measurement of The Samples

The CIELab method was employed to determine the discoloration of the samples due to smoking influence. This method involved directly measuring the values of L*, a*, and b* from a photograph. The photo was obtained using a scanner machine (CanoScan 4400F, from Canon, Japan) then analyzed with the Adobe Photoshop CS5 application [13,14]. L* indicated lightness, with a value of 0 to 100 (black to white); a* indicated colors from green to red, with +a* from 0 to 80 corresponding to red and −a* from −80 to 0 corresponding to green; and b* indicated color from blue to yellow, with +b* from 0 to 70 corresponding to yellow and −b* from −70 to 0 corresponding to blue [15]. Each sample was

assessed at five points, and the average values were used for the analysis. The color change ($\Delta E$) was calculated considering CIELab method [16], using the following equation:

$$\Delta E = \sqrt{\left[(\Delta L*)^2 + (\Delta a*)^2 + (\Delta b*)^2\right]} \tag{2}$$

where, $\Delta E$: color change; $\Delta L*$: difference in L* values between compared samples; $\Delta a*$: difference in a* values between compared samples; $\Delta b*$: difference in b* values between compared samples. The color change can be classified as shown in Table 1.

**Table 1.** Color change class [16,17].

| Class | Color Difference | Color Change Effect |
|---|---|---|
| 1 | $\Delta E < 0.2$ | Invisible changes |
| 2 | $0.2 < \Delta E < 2.0$ | Very small changes |
| 3 | $2.0 < \Delta E < 3.0$ | Small changes (color changes visible by high-quality filter) |
| 4 | $3.0 < \Delta E < 6.0$ | Medium (color changes visible by medium-quality filter) |
| 5 | $6.0 < \Delta E < 12$ | Big (distinct color changes) |
| 6 | $\Delta E > 12$ | Different color |

### 2.4. Dry-Wood Termites Resistance Test

Specimens with a 2.5 cm by 2.5 cm cross section and measuring 5 cm in the longitudinal direction were prepared according to Indonesian National Standard 7207-2014 [18]. This standard has been recognized and improved by researchers from various countries in evaluating the resistance of wood against termite as well as fungi attacks including in Japan [19]. The individual sample was placed in the center of a glass tube (3 cm high by 1.8 cm in diameter), and 50 dry-wood termite workers (*Cryptotermes cynocephalus* Light) were introduced into the glass tube. The samples were then placed in a dark room for 12 weeks. At the end of the experiment, wood specimen weight loss and termite mortality percentage were determined with five replications. The wood resistance was classified according to the Indonesian National Standard described in Table 2 [18]. The weight loss and termite mortality percentage were determined using the following equations:

$$WL\ (\%) = (W_b - W_d)/W_b \times 100\% \tag{3}$$

$$Mortality\ (\%) = (T_b - T_d)/T_b \times 100\% \tag{4}$$

where:

WL: weight loss percentage of wood specimen.
$W_b$: gewang wood weight prior to the test at oven-dried condition.
$W_d$: gewang wood weight after the test at oven-dried condition.
$T_b$: the number of live termites before the test.
$T_d$: the number of live termites after the test.

**Table 2.** Classification of wood resistance against dry-wood termites.

| Class | Sample Condition | Weight Loss (%) |
|---|---|---|
| I | Very resistant | <2.0 |
| II | Resistant | 2.0–4.3 |
| III | Moderate | 4.4–8.1 |
| IV | Poor | 8.2–28.1 |
| V | Very poor | >28.1 |

### 2.5. Fungus Decay Test

Wood decaying fungus *Schizophyllum commune* was inoculated in the glass boxes using potato dextrose agar media for 10 days to allow the fungus grew adequately [20,21]. In the next step, wood samples with a cross section of 2.5 cm × 1.5 cm and 5 cm in the longitudinal direction were placed on the fungus isolate in the glass box for the test [18]. After 10 weeks of inoculation in an incubator at a temperature ranging from 22 to 28 °C and 80 to 90 percent relative humidity, wood weight loss percentage was determined. The average MC of the wood samples prior to the test was 12 ± 3% and weight loss percentage was determined using the equation (3). The wood resistance was classified according to the Indonesian National Standard [18] described in Table 3.

**Table 3.** Classification of wood resistance against fungal decay.

| Class | Sample Condition | Weight Loss (%) |
|-------|------------------|-----------------|
| I | Very resistant | <0.5 |
| II | Resistant | 0.5–4.9 |
| III | Moderate | 5.0–9.9 |
| IV | Poor | 10.0–30.0 |
| V | Very poor | >30.0 |

### 2.6. Analysis of the Data

A 3 by 2 by 2 factorial completely randomized design was used to analyze the data where the first factor (A) is the longitudinal direction (bottom, middle, and upper part); second factor (B) is the transversal direction (inner and outer part); and third factor (C) is the treatment (unsmoked and smoked). A linear equation of mathematical model as follows:

$$Y_{ijkl} = \mu + A_j + B_k + C_l + (AB)_{jk} + (AC)_{jl} + (BC)_{kl} + (ABC)_{jkl} + \varepsilon_{ijkl} \tag{5}$$

where:

$Y_{ijkl}$: The observed value of the experimental unit *i* from the combination of *jkl* with factor A at the level to *j*, factor B at the level to *k*, and factor C at the level to *l*.

$\mu$: Grand mean parameter.

$A_j$: The influence of factor longitudinal direction at level *j* (*j* = 1, 2, 3).

$B_k$: The influence of factor transversal direction at level *k* (*k* = 1, 2).

$C_l$: The influence of factor treatment at level *l* (*l* = 1, 2).

$AB_{jk}$: The influence interaction between factor A at the level *j* and factor B at level to *k*.

$AC_{jl}$: The influence interaction between factor A at the level *j* and factor C at level to *l*.

$BC_{kl}$: The influence interaction between factor B at the level *k* and factor C at level to *l*.

$ABC_{jkl}$: The influence interaction between factor A at the level *j*, factor B at level *k*, and factor C at the level *l*.

$\varepsilon_{ijkl}$: The effect of the error that arises the *i*-th from experiment combination in factor A at level *j*, factor B at level *k*, and factor C at level *l*.

Duncan's multiple range tests were used for further analysis if a factor was significantly different at $p < 0.05$.

## 3. Results and Discussion

### 3.1. Density of Gewang Wood

It seems that smoke treatment did not affect the density of the samples because the precipitated smoke on the wood surface was very small which was not sufficient to influence the density increment [22]. The density values of unsmoked samples are illustrated in Figure 3.

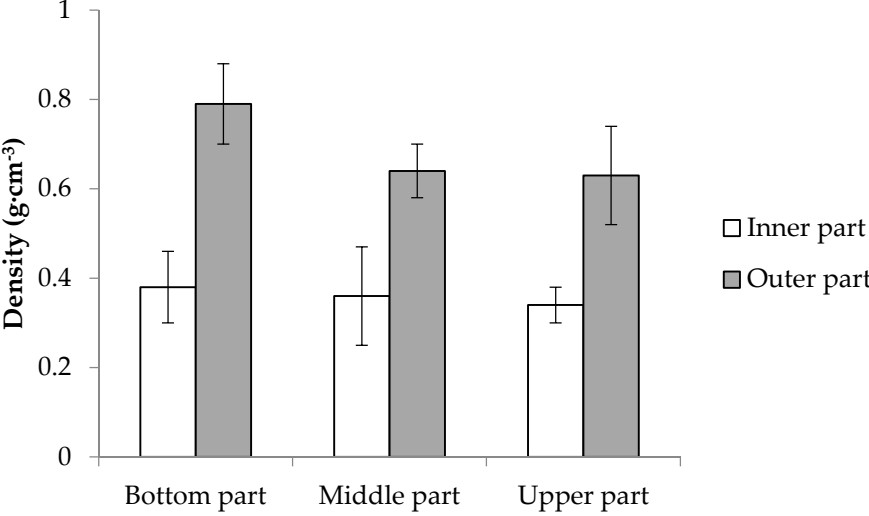

**Figure 3.** Density of unsmoked samples.

The summary of variance analysis regarding the effect of longitudinal and transversal parts on density of the samples is shown in Table 4. It could be mentioned that the density was significantly influenced by the longitudinal and transversal directions, and the interaction between them was not significant. It appears that outer part (average $0.69 \pm 0.09$ g·cm$^{-3}$) had a higher density compared to the inner part (average $0.36 \pm 0.08$ g·cm$^{-3}$) as shown in Figure 3. A decrease in the density from bottom towards the upper part could also be responsible for such findings. This could be due to the proportion of vascular bundles within the gewang trunk as well as the other palmae species which is decreasing from the bottom part to the upper part of the trunk, and the same phenomenon was also observed from the outer part to the inner part, as described in Figure 4 [23]. The lower proportion of the vascular bundles caused the lower density of the wood [5]. This finding was in line with Subyakto et al. [4] who found that the outer part of gewang trunk has a higher density compared that of the inner part of the trunk.

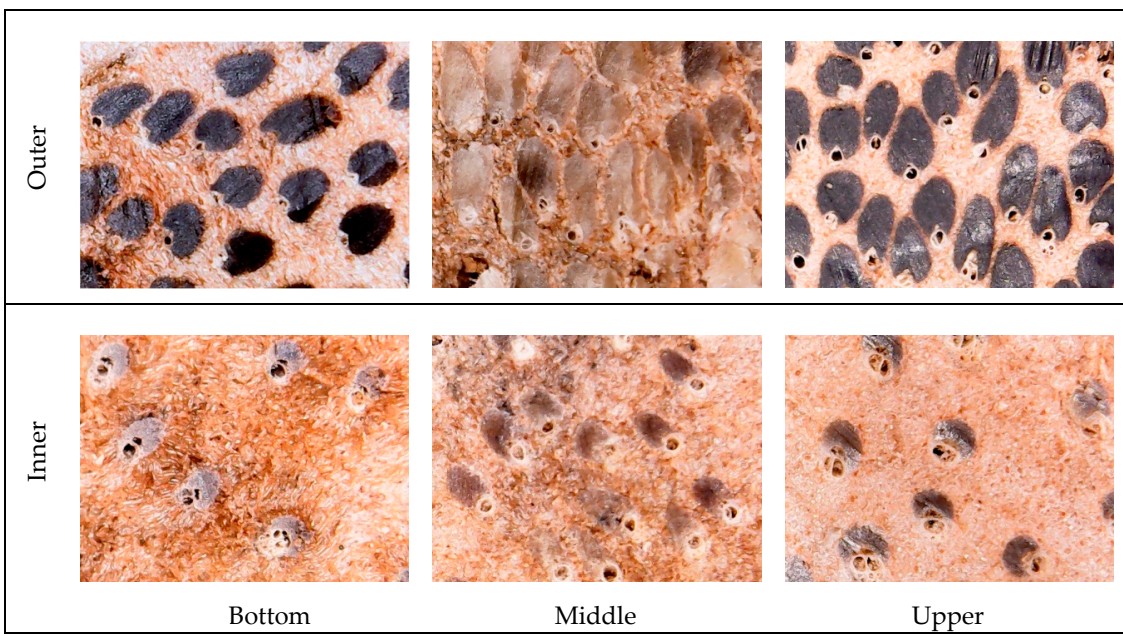

**Figure 4.** Macroscopic images (20×) of gewang wood.

**Table 4.** Variance analysis summary of density, termite mortality, weight loss on termite test, and weight loss on fungal test.

| Parameter | Density | Termite Mortality | WL Termite | WL Fungi |
|---|---|---|---|---|
| Longitudinal Part (A) | * | Ns | ** | ** |
| Transversal Part (B) | ** | * | ** | ** |
| Treatment (C) | - | ** | ** | ** |
| A × B | Ns | Ns | Ns | Ns |
| A × C | - | Ns | Ns | ** |
| B × C | - | * | ** | ** |
| A × B × C | - | Ns | * | Ns |

** Highly significant difference ($p \leq 0.01$); * = significantly difference ($p \leq 0.05$); Ns = not significantly different ($p \leq 0.05$).

## 3.2. Discoloration of the Samples

The wood color was indicated by values of L* (lightness, black to white), a* (redness, green to red), b* (yellowness, blue to yellow), and ΔE for the color change, which is shown in Table 5. The color of the specimens was shown in Figure 5.

**Table 5.** Color of unsmoked and smoked wood specimens.

| Treatment | L* | a* | b* | ΔE |
|---|---|---|---|---|
| Unsmoked | 50.5 (4.7) | 8.3 (2.5) | 22.9 (1.5) | |
| Smoked | 25.0 (2.0) | 6.5 (1.8) | 8.4 (1.5) | 29.7 (3.7) |
| t-test | 0.00 ** | 0.34 ns | 0.00 ** | |

** Highly significant difference ($p \leq 0.01$); ns = not significantly different ($p \leq 0.05$), (numbers in parentheses are standard deviation values).

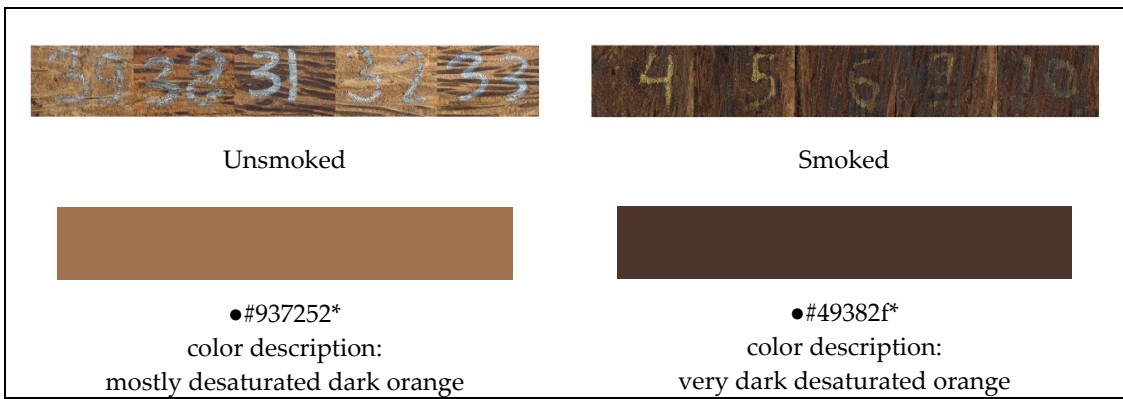

Unsmoked

•#937252*
color description:
mostly desaturated dark orange

Smoked

•#49382f*
color description:
very dark desaturated orange

**Figure 5.** Discoloration of unsmoked and smoked specimens. * Remarks: the color description defined by ColorHexa [24].

Based on the results displayed in Table 5, unsmoked and smoked samples were significantly different from each other in terms of their L* and b* values. Smoke treatment resulted in samples with a darker color, as indicated by an L* value of smoked wood that decreased to become half of the value of the unsmoked wood. Smoke treatment changed the b* value blue to yellow color, and the average values declined or became more blue color. In terms of a* value green to red color, unsmoked and smoked were not significantly different, as indicated by the value changed was a small number. These findings revealed that smoke treatment for samples resulted in different colors (ΔE more than 12) compared with unsmoked samples. In a previous work, similar results for mangium and sengon smoked woods were determined [12,22], Ishiguri et al. [25] also determined that wood samples treated with smoke for 200 h showed significant discoloration having deeper color those of exposed of smoke for 100 h.

### 3.3. Resistance of the Samples to Dry-Wood Termite

Termite mortalities of treated specimens are displayed in Table 6, and analysis of variance is also presented in Table 4. Regarding to analysis of variance, the factors of transversal part (direction), treatment, and interaction of the both factors significantly affected termite mortality ($p < 0.05$). In terms of transversal direction, the outer part had higher mortality than the inner part, because the outer part had a much higher density than the inner part. Such a finding was in line with Hadi et al. [26] who found that untreated wood with a lower density (sugi, *Cryptomeria japonica,* with density 0.34 g·cm$^{-3}$) had a lower dry-wood termite mortality compared to higher density wood (mindi, *Melia azedarach,* with density 0.43 g·cm$^{-3}$).

**Table 6.** Mortality rate of dry-wood termite.

| Trunk Part | Inner Part (%) | | Outer Part (%) | |
|---|---|---|---|---|
| | Unsmoked | Smoked | Unsmoked | Smoked |
| Bottom part | 66.2 (10.0) | 100 (0) | 71.2 (3.9) | 100 (0) |
| Middle part | 65.2 (10.7) | 100 (0) | 70.4 (1.7) | 100 (0) |
| Upper part | 63.6 (5.2) | 100 (0) | 68.4 (3.6) | 100 (0) |

Numbers in parentheses are standard deviation values.

Regarding the smoke treatment of the samples, termite mortality of control samples ranged from 63.6 to 71.2%, while smoked specimens had complete termite mortality. In the case of kesambi samples exposed to smoke, acetic acid was produced, followed by phenol, ketones, amines and benzene, and such chemicals created a toxic environment for the termite [22]. These chemicals also could precipitate on the samples, resulting in their discoloration. Such a conclusion was observed in two of the past studies carried out Hadi et al. [27,28]. Abdillah et al. [29] found that wood smoke could penetrate the low and medium density wood (jabon wood and pine wood) until 0.69 and 0.74 mm, respectively, in smoking treatment.

Weight loss percentage value and resistance class as well as variance analysis of dry-wood termite are displayed in Figure 6 and Table 4, respectively. The longitudinal part, transversal part, treatment, and interaction of the three factors significantly influenced the percentage wood weight loss ($p < 0.05$). In terms of longitudinal direction, the wood from bottom part had a lowest weight loss and the values increased toward upper parts. Such results were in line with the termite mortality and the correlation could be explained by the fact that the highest termite mortality would have the lowest wood weight loss, and vice versa.

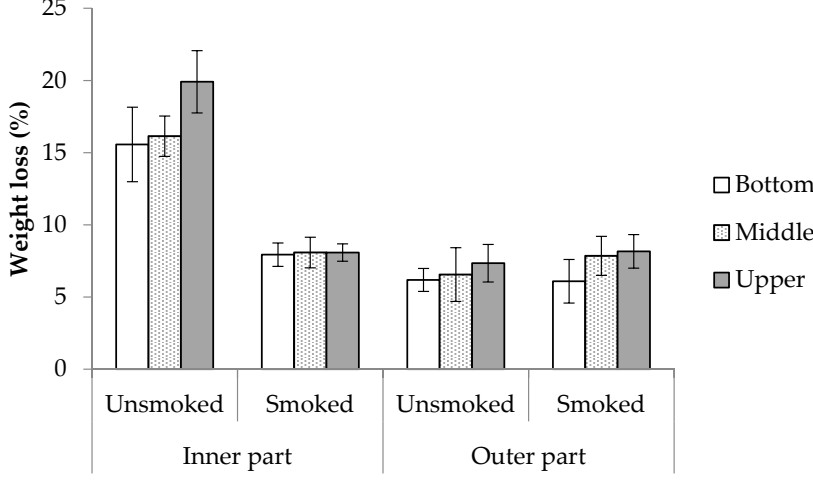

**Figure 6.** Wood weight loss after 12 weeks exposed to dry-wood termite.

Regarding to the transversal direction, the samples taken from the inner part of trunks had a higher weight loss compared to those taken from the outer part. In other words, the inner part was more susceptible to dry-wood termite attack, because the inner part had a lower density than the outer part. These findings were in line with Arango et al. [30] who analyzed six hardwood species, and found a significant inverse association between percentage of mass lost and specific gravity; in other words, wood with a higher specific gravity has more resistance to *Reticulitermes flavipes* Kollar termites.

Another factor of treatment, exposed smoke on the inner part could enhance the resistance of gewang wood to dry-wood termite attack, as indicated by smoked wood having a much lower percentage wood weight loss compared to unsmoked samples. Furthermore, when considering the resistance classes in Table 7, unsmoked samples on the inner part of gewang wood belonged to class IV (poor resistance to dry-wood termite attack), but the smoked wood of inner part belonged to class III (moderately resistant). In other words, the smoked gewang wood of inner part had one class higher than unsmoked samples. Meanwhile, the exposed smoke did not affect the durability of the outer part of gewang wood, since this part already had moderate resistance to dry-wood termite attack (class III, the same class as the inner part of the smoked wood), and it had the utmost level of resistance against termite attack. This finding was in line with the findings of previous studies determined that smoke treatment could enhance the resistant of low density wood to be equal to high density smoked wood at the highest level of termite resistance in particular circumstances [12,27,28].

**Table 7.** Resistance class after exposed to dry-wood termite.

| Trunk Part | Inner Part | | Outer Part | |
|:---:|:---:|:---:|:---:|:---:|
| | Unsmoked | Smoked | Unsmoked | Smoked |
| Bottom | IV | III | III | III |
| Middle | IV | III | III | III |
| Upper | IV | III | III | III |

### 3.4. Fungal Decay Assay Results

The weight loss value and analysis of variance of fungal test are shown in Figure 7 and Table 4. The percentage weight loss of the fungus test ranged from 4.86 to 18.12% for unsmoked samples. Smoked samples had a lower weight loss than those unsmoked specimens. Analysis of variance revealed that each factor had a highly significantly different ($p < 0.01$), but there was no interaction between the longitudinal with transversal part of the gewang trunk and three factors. Regarding the resistance class in Table 8, smoke treatment increased the resistance to fungus decay of the samples taken from the inner and outer part of the trunk. Smoking process enhanced the resistance of the samples class to be two classes higher. Additionally, smoke treatment on the samples taken from outer and inner parts had the same resistance class, according to SNI 2014 [18].

It is known that smoke has toxic or carcinogenic compounds such as the polycyclic aromatic hydrocarbons (PAH) group that can cause permanent damage to living organisms [31,32]. The phenol group, as a constituent of smoke, can be used as a wood preservative [11,12,33]. These components were not only increasing termite mortality [33], but also can be considered as an environmentally friendly approach to enhance fungus resistance of the samples [34].

**Table 8.** Wood resistance class after exposure to decay fungi.

| Trunk Part | Inner Part | | Outer Part | |
|:---:|:---:|:---:|:---:|:---:|
| | Unsmoked | Smoked | Unsmoked | Smoked |
| Bottom | IV | II | II | II |
| Middle | IV | II | III | II |
| Upper | IV | II | III | II |

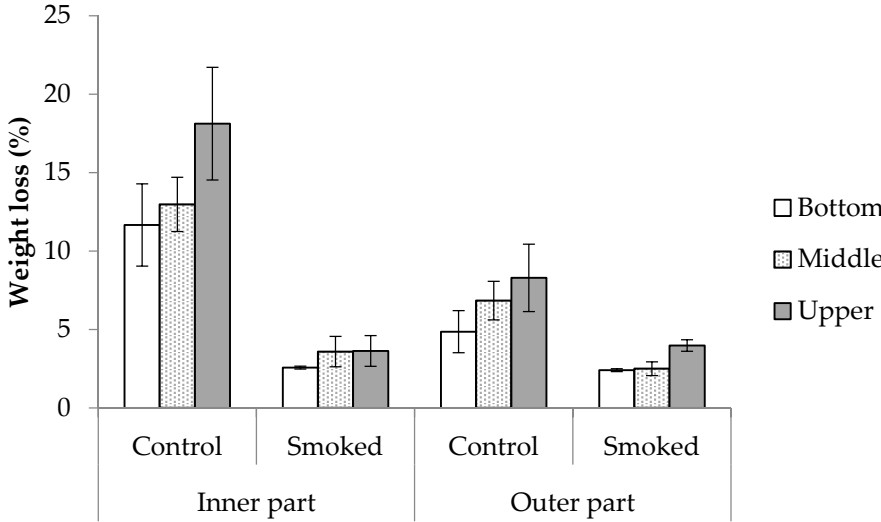

**Figure 7.** Wood weight loss after 10 weeks exposed to decay fungus.

## 4. Conclusions

Based on our findings, the following conclusions can be drawn:

a.   The inner part and upper part of wood from the gewang trunk had lower densities than the outer part and bottom part, respectively.

b.   The color of smoked gewang wood differed in color from unsmoked samples.

c.   Unsmoked samples of gewang wood from the inner part were more susceptible to attack by dry-wood termite and fungal attacks than the outer part.

d.   Smoke treatment was very effective for the inner part, enhancing resistance to dry-wood termite and fungal attacks.

**Author Contributions:** Conceptualization, D.N., W.D.; Data Curation, I.B.A., W.D., L.K.; Investigation Resources, Y.S.H. and S.H.; Writing-review and editing, D.N., Y.S.H., S.H., I.B.A. All authors have read and agreed to the published version of the manuscript.

**Funding:** Indonesian Ministry of Education and Culture through Priority Basic Research of the University Grant (Penelitian Dasar Unggulan Perguruan Tinggi) Year 2020.

**Acknowledgments:** This research was a part of Basic Research Year 2020 granted by the Deputy of Research and Development Strengthening, Ministry of Research and Technology, the Republic of Indonesia.

**Conflicts of Interest:** The authors declare no conflict of interest.

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
