# Peer review of "Evaluation of Color Change and Biodeterioration Resistance of Gewang (Corypha utan Lamk.) Wood"

_applsci, doi:10.3390/app10217501_

Round 1

Reviewer 1 Report

Abstract

Please this section needs more attention, specifically when it comes to the results. Rephrase this sentence in line 24-25, “so that such species can be used for building application with a better more efficient utilization”.

Introduction

In my opinion the information found on the second paragraph is irrelevant to the objective of this study.

L74 - compared (delete each other)

Materials and methods

Figure 1.

I have some problems with this picture, for some reason it is blurry and needs major improvements. I recommend the authors to update the picture. Also the caption should be right after the figure and not between the text.

L105 - I do not know if I understood here, were 18°C initial and 36°C final temperature? or the chamber had variations going from 18 to 36 °C. Please clarify this information.

L112-124 How did the authors measure color using a photoshop? I have never seen color being measure that way. Why did you measure color this way? First, CIELab method describes color variations that were measured using a spectrophotometer. In what did you base your measurements? Have you seen any other work that measure color change like that? If so, could you reference that? This section needs to be clearer.

L148-154 I understood that the authors performed a decay test. However, several important details are missing here. Were the samples ovendried before test? How long did the fungus take to “grow adequately”? How many weeks before the authors exposed the samples?  Were the glass boxes placed in an incubator?

L158-162 This section needs rearranging and rewriting.

Results and discussion

The results need to be discussed. For example, why did smoke treatment work only for inner part samples?

L177 Why is that?

“outer part of gewang trunk has a higher density compared to the inner part of the trunk”.

L211-218 Please check the English here; it really needs improvement.

Author Response

Dear Reviewer,

Please see the file for detailed revision for comment and revision.

Best regards,

Authors

Reviewer 2 Report

This paper attempts to show evaluation of color change and biodeterioration resistance of Gewang (Corypha Utan Lamk.) wood. The study drew useful results by some experiments. This paper addresses an important topic, but it still needs major revisions as followings:

  1. First of all, the authors should clearly explain in the text what the scholarly significance of this manuscript is. At present, it seems to be just a technical report, but not an academic paper.
  2. Since this paper deals with natural materials, the authors need to explain in the text what they think about the variability of the results (differences between individuals). It is recommended that the data will provide a convincing explanation if possible.
  3. It is understandable that the evaluation standard is an Indonesian standard, but it is necessary to explain in the text the international position of the standard.
  4. Most of the results of the experiments are shown in tables, but some effort should be made to deepen the reader's understanding by using figures such as graphs.

Author Response

(The authors gave the same response as above.)

Reviewer 3 Report

Dear Authors

This paper is interesting in making wood resistant to fungi and termite. However, ti would be good to know the thickness of wood dependij go the levels of smoke treatments.

As well known, the charred wood is resistance to termite and fungi because there is no nutrients in the hcarrec wood, which has been knwon for time in handling wood. However, it would be good to make it more scientific anc understandable in terms of wood science.

Sincerely 

Author Response

(The authors gave the same response as above.)

Round 2

Reviewer 1 Report

Dear authors

The paper had major improvements after revision. Great job!

Reviewer 2 Report

Nothing.